# Membrane Association of the Short Transglutaminase Type 2 Splice Variant (TG2-S) Modulates Cisplatin Resistance in a Human Hepatocellular Carcinoma (HepG2) Cell Line

Dipak D. Meshram [1,2], Cristina Fanutti [1], Claire V. S. Pike [1] and Peter J. Coussons [1,*]

[1] Cancer Cell Biology Subgroup, Biomedical Research Group, School of Life Sciences, Faculty of Science and Engineering, Anglia Ruskin University, Cambridge CB1 1PT, UK; d.meshram@sheffield.ac.uk (D.D.M.); c.fanutti@gmail.com (C.F.); claire.pike@aru.ac.uk (C.V.S.P.)

[2] School of Biosciences, The University of Sheffield, Firth Court, Western Bank, Sheffield S10 2TN, UK

[*] Correspondence: peter.coussons@aru.ac.uk

**Abstract:** Hepatocellular carcinoma (HCC) is a heterogeneous malignancy with complex carcinogenesis. Although there has been significant progress in the treatment of HCC over the past decades, drug resistance to chemotherapy remains a major obstacle in its successful management. In this study, we were able to reduce chemoresistance in cisplatin-resistant HepG2 cells by either silencing the expression of transglutaminase type 2 (TG2) using siRNA or by the pre-treatment of cells with the TG2 enzyme inhibitor cystamine. Further analysis revealed that, whereas the full-length TG2 isoform (TG2-L) was almost completely cytoplasmic in its distribution, the majority of the short TG2 isoform (TG2-S) was membrane-associated in both parental and chemoresistant HepG2 cells. Following the induction of cisplatin toxicity in non-chemoresistant parental cells, TG2-S, together with cisplatin, quickly relocated to the cytosolic fraction. Conversely, no cytosolic relocalisation of TG2-S or nuclear accumulation cisplatin was observed, following the identical treatment of chemoresistant cells, where TG2-S remained predominantly membrane-associated. This suggests that the deficient subcellular relocalisation of TG2-S from membranous structures into the cytoplasm may limit the apoptic response to cisplatin toxicity in chemoresistant cells. Structural analysis of TG2 revealed the presence of binding motifs for interaction of TG2-S with the membrane scaffold protein LC3/LC3 homologue that could contribute to a novel mechanism of chemotherapeutic resistance in HepG2 cells

**Keywords:** hepatocellular carcinoma (HCC); HepG2; transglutaminase 2 (TG2); cisplatin; chemoresistance; sub-cellular localisation; apoptosis; autophagy

## 1. Introduction

Hepatocellular carcinoma (HCC) is the most common form of liver cancer (90%) [1–3], having the shortest survival time (<6 months) [4]. Its poor prognosis is mainly due to late diagnosis, which reduces chances of survival, and to the relatively high cost and lack of availability of effective treatments, particularly in developing countries.

Current therapies for liver cancer include the following three major types of cancer treatment: surgical resection, radiation therapy, and drug therapy. Drug therapy is further divided into chemotherapy, molecular targeted therapy, and immunotherapy. Chemotherapy is used solely or in combination with surgical resection, radiation therapy, or other drug therapies, depending on the type and progression of the cancer [5–7].

The platinum-based antineoplastic drug cisplatin is still being used for the treatment of advanced hepatocellular carcinoma, as the sole chemotherapy drug or in combination with other drugs [8,9]. Although it is a well-established, safe, and cost-effective treatment for patients, cisplatin has clinical drawbacks associated with its relatively high toxicity and the rapid development of chemoresistance that is seen in many tumours [10]. Wide differences in its effectiveness also stem from the variety of clinical approaches used for drug

delivery [11–13]. Thus, the optimisation of cisplatin efficiency [14,15] and the minimisation of cisplatin chemoresistance [16] are the most important approaches that could improve patient treatment and prognosis.

"Tissue" or type 2 transglutaminase (often described as "liver transglutaminase", due to its high abundance in hepatic tissues), is a ubiquitous and pleiotropic enzyme that is overexpressed in many cancer cells, where it constitutively activates the stress-responsive transcription factor nuclear factor-kappaB [17]. Its calcium-dependent activity results in the synthesis of glutamyl-lysyl isopeptide cross-links and/or a range of post-translational modifications of proteins, including deamidation and amine incorporation into reactive glutaminyl side-chains of protein substrates [18–20]. This activity can inhibit or stimulate diverse processes, such as apoptic body formation, cell adhesion, and endocytosis [21–23], thus modulating cell growth and death, tumour metastasis, and drug resistance [24–27].

The alternative splicing of TG2 RNA transcripts produces several protein TG2 iso-forms [28–35], of which the 'long' TG2-L (78 kDa) and the 'short' TG2-S (61 kDa) isoforms are the most commonly expressed. This difference in molecular weight is due to a TG2-L-associated GTP-binding domain, which is absent in TG2-S. Thus, TG2-L can legitimately be termed a 'small' calcium-dependent G-protein that acts as a molecular switch inside cells. Such proteins typically transduce signals from a variety of stimuli, often from outside cells to their interiors and are, in some cases, established modulators of endocytosis [36–38].

Phatak et al. (2013) [32] showed differences in alternative splice variants of TG2 in cancer cell lines, suggesting that the alternative splicing of TG2 is a more active process in cancer cells than in normal cells. Thus, it may be the case that TG2-related effects are not simply due to altered cellular levels of transglutaminase protein expression or associated changes in transaminating enzyme activity per se, but that chemoresistance may also be influenced by the differential expression of TG2 isoforms. Such questions are relevant because, depending upon the levels of expression of each splice variant and the local levels of intracellular calcium and GTP, TG2 may exhibit large changes in its site-specific cellular activity, which characterise the highly pleiotropic nature of TG2 gene expression in general.

Under normal conditions, when intracellular calcium levels are low (100–200 nM) and GTP levels are high (0.1–1 mM), TG2-L's relatively high Km (1–5 μM) for calcium [39] is thought to reflect its 'closed' and enzymically inactive conformation, allowing it to function simply as an accessory adhesion protein that is protective against cellular stress. Structural studies [40–43] support the notion that TG2-L only adopts an 'open' structure and becomes enzymically active under conditions where the intracellular GTP levels drop and/or those of calcium rise, e.g., as occurs when cells become severely stressed or undergo apoptosis. Conversely, TG2-S is thought to adopt a relatively "open" conformation, becoming the constitutive transamidation-active form of TG2. Whilst the Janus-like interplay between TG2-L and TG2-S fits neatly into a binary mechanistic model, where TG2-L promotes cell survival and its GTP-insensitive and enzymically competent TG2-S counterpart promotes cytotoxicity-related cell death, this model does not fully explain the role of TG2-S in the development of chemoresistance [44].

Here, we show that the treatment of HepG2 cells with cisplatin causes a redistribution of TG2-S from a membrane-associated assembly to a cytoplasmic pool, an effect that was absent in cells that had been previously made chemoresistant to cisplatin. On this basis, we propose a model whereby changes in the sub-cellular localisation of the short TG2 variant (TG2-S) contribute to cisplatin resistance in human hepatocellular carcinoma cells, thus reducing cisplatin toxicity and promoting cell survival and chemoresistance in HepG2 cells.

## 2. Materials and Methods

All chemicals were from Sigma Aldrich, Gillingham, UK, unless otherwise stated. To identify the differences in the accumulation or uptake of cisplatin in cisplatin-resistant HepG2/cr and sensitive parental HepG2 cells, we used the commercially available fluorescent cisplatin-Alexa fluor 546-complex, which has the same DNA-binding mechanism

as clinically used cisplatin (ULYSIS Nucleic acid labelling kit, Molecular probes, Thermo Scientific, Swindon, UK).

### 2.1. Preparation of Stock Solutions

Stock solutions of 0.5 M cystamine and 2 mM cisplatin were freshly prepared in 18 m$\Omega$ ultrapure water and were stored under subdued light at 4 °C and room temperature, respectively, prior to use.

### 2.2. Cell lines and Cell Culture Preparation and Maintenance

The HepG2 cell line used in our experiments was from the European Collection of Cell Cultures (ECCCs) and was cultured in RPMI 1640 culture medium (Invitrogen, Paisley, UK), fully supplemented with 10% (*v/v*) heat-inactivated foetal calf serum (FCS), 2 mM L-glutamine, 100 U/mL penicillin, and 100 µg/mL streptomycin. Cell cultures were maintained at 37 °C, in the presence of a humidified 5% $CO_2$ atmosphere, and were maintained as being free of mycoplasma contamination by adding Plasmocin, a commercially available antibiotic. The healthy condition of cells was checked daily using light microscopy and their morphological authentication was confirmed by comparison with STR profile data that are available online (NCBI Biosample database).

### 2.3. Cell Cytotoxicity Measurements Using the Cell Counting Kit-8 (CCK-8) Assay

Cells were trypsinised (0.1% trypsin/0.02% EDTA) and re-suspended in fresh culture medium at a concentration of $1 \times 10^5$ cells/mL and were then seeded into 96-well cell culture plates at a density of 10,000 cells per well, in 100 µL of culture medium. Following 24 h incubation, the culture medium was removed and replaced with 100 µL of fresh culture medium containing cisplatin or the other drugs tested. Following incubation, cell viabilities were measured using CCK-8 cell counting kits, according to the manufacturer's instructions (for protocol details see Meshram et al., 2017) [45]. Cellular viabilities were calculated on the basis of $IC_{50}$ values, which are defined as the concentration of toxic agent required to reduce cell survival by 50%. In each case, the absolute $IC_{50}$ was calculated using non-linear regression (curve fit) analysis, where the values of the x and y axes represent concentration and response, respectively.

### 2.4. Single Dose Treatment for the Development of Cisplatin-Resistant HepG2cr Cells

In order to develop cisplatin-resistant cell lines, cisplatin toxicity curves were constructed using the CCK-8 Cell Counting Kit-8 assay (Section 2.3). Cisplatin-resistant HepG2 cells were developed following a single dose treatment, according to the conditions previously described [46,47], with minor modifications (described below). Parental and HepG2/cr cells were then grown at 37 °C in a humidified 5% $CO_2$ atmosphere for 24 h, at a density of $1 \times 10^6$ cells per 75 cm$^2$ culture flask. Duplicate samples of cells were treated with 8 µM cisplatin for 4 days. Following treatment, cells were washed twice with warm culture medium, before being incubated in cisplatin-free culture medium for a further 4 weeks; fresh medium was supplied every 2–3 days. The surviving colonies were recovered by passage before being frozen down in liquid nitrogen. The control flasks were treated identically, except for the omission of cisplatin. The cell colonies collected from cisplatin-treated flasks were named HepG2/cr cells and the morphology of both cell lines was monitored using inverted phase contrast microscopy. Subsequently, cells were exposed to 0–20 µM of cisplatin for 24 h or 48 h, before the drug sensitivity of HepG2/cr cells was compared with that of non-drug-resistant parental control cells.

### 2.5. Cell Migration Assay

Approximately 400,000 cells/well were grown in 12-well culture plates for 24 h. Then, the medium was removed and a horizontal scratch wound was created across the monolayer using a sterile 10 µL pipette tip. The remaining adherent cells were washed twice with fresh medium to remove detached cells. Adherent cells were incubated at 37 °C

in a 5% $CO_2$ atmosphere, before cellular regrowth was measured using inverted phase contrast microscopy.

### 2.6. Inhibition of TG2 by Cystamine

Both parental HepG2 and cisplatin-resistant HepG2/cr cells were incubated in the presence or absence of non-toxic 2 mM concentrations of cystamine for 48 h. Then, cell lysates were prepared using RIPA buffer and the TG2 enzyme activity was measured using a TG2-specific activity assay kit (CovaLab, Bron, France).

### 2.7. siRNA Knockdown Inhibition of TG2 Expression

Silencing RNA (siRNA) knockdown studies employed siRNA oligomer sequences, which had been pre-designed and synthesised by Ambion, Life Technologies, Paisley, UK, to inhibit TG2 expression. The sequences used are as follows: antisense-UCUUAGUGGAAAA CGGGCCtT; sense-GGCCCGUUUUCCACUAAGATT. Cells were transfected with oligomers using Lipofectamine 2000 transfection reagent (Life Technologies, UK), according to the manufacturer's instructions.

### 2.8. Reverse Transcription Quantitative Real-Time PCR (RT-qPCR)

In order to allow cells to attach to the bottom of the flasks, HepG2 and HepG2/cr cells were seeded at $\geq 1 \times 10^6$ cells per 25 cm$^2$ culture flask and were then pre-incubated under standard conditions (Section 4.2) for 24 h. Following subsequent treatments, with or without cisplatin, total RNA was isolated from cells using the RNeasy kit (Quiagen, Manchester, UK), according to the manufacturer's instructions. Following DNase treatment, total RNA was collected in RNase-free $H_2O$ and was quantified using a Nanodrop spectrophotometer (Thermo Scientific, UK).

The quantitative estimation of the expression of TGM2 or other genes was based on the real-time monitoring of the amplification and melting curves. The reverse transcription of RNA was performed with a 500 ng final concentration of total RNA, using the iScript reverse transcriptase for a one-step RT-PCR kit (Bio-Rad, Watford, UK), with modifications, as described by the manufacturer's protocol and specific reverse primers (Eurofins, Guildford, UK). The primer sequences used for the RT-PCR of GAPDH and TG2 are shown in Table 1. For the full detailed protocol, see Meshram et al., 2017 [45].

**Table 1.** Primer sequences used for the RT-PCR of GAPDH and TG2 (F = forward primer, R = reverse primer).

| Transcript Name | Sequence |
| --- | --- |
| GAPDH | (F)—5′-CACTAGGCGCTCACTGTTCTC-3′<br>(R)—3′-GACTCCACGACGTACTCAGC-5′ |
| TG2 | (F)—5′-CTGGGCCACTTCATTTTGC-3′<br>(R)—3′-ACTCCTGCCGCTCCTCTTC-5′ |

### 2.9. Isolation of Cell Membrane Proteins

Cell membranes were isolated from both cisplatin-treated HepG2 and HepG2/cr cells and from untreated controls, using a Mem-PER Plus membrane protein extraction kit (Thermo Scientific, UK), according to the manufacturer's instructions.

### 2.10. Measurement of TG2 Isoform Expression using Western Blotting

Cell lysates were prepared using RIPA cell lysis buffer (50 mM Tris-HCl, pH 8.0; 150 mM NaCl; 1% NP-40; 0.5% sodium deoxycholate; and 0.1% SDS) with a freshly added 1% complete protease inhibitor cocktail (aprotinin, bestatin, leupeptin, and pepstatin A), unless otherwise described. Samples were separated using SDS-PAGE and were electrophoretically transferred onto nitrocellulose membranes by semi-dry transfer. Blots were blocked in a 5% solution of non-fat dried milk powder for 1 h, incubated with primary

following secondary antibodies, and developed using HRP substrate [45]. The primary antibody used was mouse anti-TG2 antibody (cat. no. ab2386, Abcam, Cambridge, UK). This antibody recognises both TG2-L and TG2-S isoforms, as confirmed in our study.

### 2.11. In Vitro Specific TG2 Colorimetric Microassay (TG2-CovTest)

Cell lysates were prepared from cisplatin-treated and non-treated HepG2 cells and were then assayed for TG2 activity, using a commercially available specific TG2 colorimetric microassay kit (Covalab, France), according to the manufacturer's instructions; see Meshram et al. (2017) [45] for the detailed protocol.

### 2.12. Confocal Microscopic Detection of Uptake of Alexa Fluor 546-Labelled Cisplatin

HepG2 and HepG2/cr cells were grown overnight on 4-well glass Lab-Tek chamber slides (Nalgen Nunc International, Rochester, NY, USA). They were then incubated with Alexa Fluor 546-labelled cisplatin (F-cisplatin; Molecular probes, Life Technologies, UK) at a final concentration of 200 U/mL (1 unit is defined as the reagent solution required to label 25 ng of DNA in vitro at 37 °C, for up to 48 h). The cells were then washed again with ice-cold PBS and fixed with ice-cold 70% ethanol for 15 min at −20 °C, before being washed again with ice-cold PBS. Cover slips were then mounted atop, using fluorescence mounting medium (Sigma Aldrich, UK), before being analysed at ×400 magnification, using an LSM510 laser-scanning microscope (Zeiss, Heidelberg, Germany).

### 2.13. Alexa Fluor 546-Labelled Cisplatin Uptake Assay using Flow Cytometry

Approximately $5 \times 10^5$ cells were grown for 24 h, to approximately 90% confluence, in 6-well culture plates, before being exposed to Alexa fluor 546-labelled cisplatin at a final concentration of 40 U/mL for 0–2 h. Cells were then collected by trypsinisation, washed with ice-cold PBS, and re-suspended in 300 μL ice-cold PBS. Finally, fields of 10,000 cells were analysed immediately using BDAccuri C6 flow cytometry (BD Biosciences, Berkshire, UK), with the fluorescence intensity detected using a laser beam set at 488 nm and a band pass filter FL-2 at 585/40.

### 2.14. Statistical Analysis

All statistical analyses were performed using GraphPad Prism (version 10.2.2) software (GraphPad, USA). Student's *t* test and one-way or two-way ANOVA with Sidak's multiple comparison tests were used to calculate significant values. *p* values < 0.05 were considered statistically significant. Significances are represented in the figures as follows: ns, $p > 0.05$; * $p < 0.05$; ** $p < 0.01$; *** $p < 0.001$; **** $p < 0.0001$, unless individual *p* values are stated. Fiji (ImageJ2.14.0/1.54f) (NIH, Bethesda, MD, USA) was used for cell migration quantification and confocal image analysis.

### 2.15. Structure Analysis and Molecular Docking Simulations

The TG2 protein sequence (UniProt accession n. P21980) was scanned for potential functional sites using the Eukaryotic Linear Motif (ELM) prediction webserver. The three-dimensional structure prediction model of TG2-S (residues 1–548) was generated using PHYRE2 Protein Fold Recognition Webserver. The protein–protein docking simulation was performed using the High Ambiguity Driven protein–protein DOCKing (HADDOCK version 2.2) Webserver [48]. The PDB accession numbers of TG2 and LC3 were 3ly6 and 1ugm, respectively. Protein–protein interaction was analysed using the PDBsum Webserver (EMBL-EBI, Cambridge, UK). All the rendering and 3D structure visualisations were performed using the PyMOL (version 2.6.0) molecular visualisation system, distributed by Schrödinger, New York, NY, USA.

## 3. Results

### 3.1. Morphological and Phenotypic Characterisation of HepG2 and HepG2/cr Cells

Treatment with cisplatin for four days killed approximately 90% of the original cells that were seeded on plates, leaving approximately 10% viable cells remaining. When observed using inverted phase contrast microscopy, the residual cisplatin-resistant cells (HepG2/cr) differed in their morphology and growth features compared to HepG2 parental cells (Figure 1A–C).

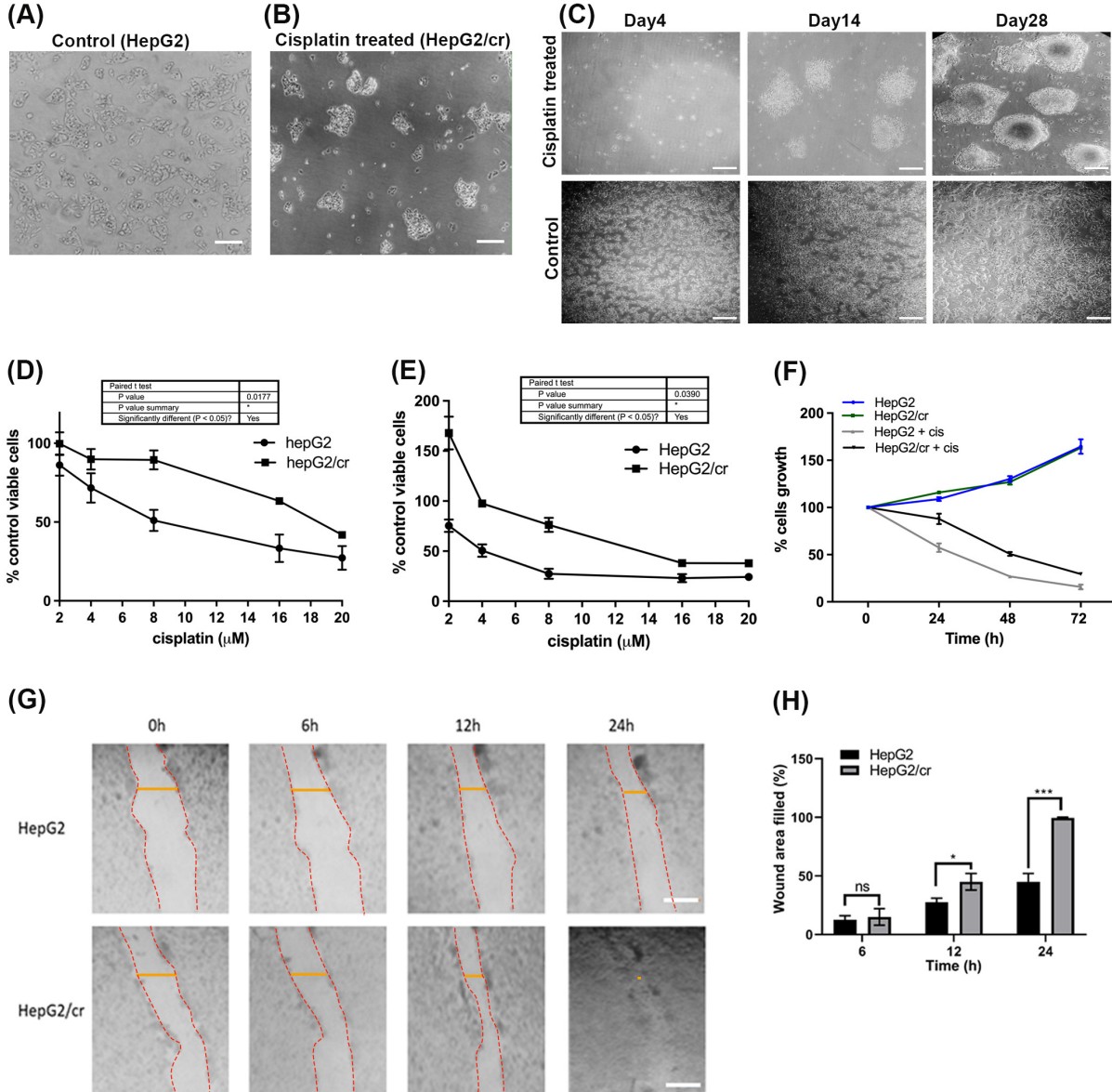

**Figure 1.** Characterisation of the cisplatin-resistant HepG2 (HepG2/cr) cell line. HepG2 cells were treated with a single dose of cisplatin (8 μM) for 4 days. Following this treatment, cells were incubated in cisplatin-free media for up to a further 4 weeks, during which time, cell morphologies were observed. Upper panel (**A**): Parental HepG2 cells before treatment (scale bar = 100 μm); (**B**): HepG2/cr recovered from cisplatin-resistant colonies (scale bar = 100 μm). (**C**): Upper panels: HepG2/cr cells incubated in cisplatin-free media for 4 days (left panel), 14 days (central panel), and 28 days (right panel), following 4-day exposure to cisplatin. Lower panel: untreated control HepG2 cells incubated in cisplatin-free media for 4 days (left panel), 14 days (central panel), and

28 days (right panel). (Scale bar = 200 μm). (**D,E**): Determination of cisplatin sensitivity in parental HepG2 and chemoresistant HepG2/cr cells. Drug-resistant phenotypes were characterised by incubating cells with 0–20 μM cisplatin for 24 h (**D**) and 48 h (**E**). The 100% cell viability is counted at 0 μM of cisplatin on the x axis. Error bars are presented as mean ± SEM of at least three independent determinations and *p* values are presented as * *p* < 0.05. (**F**) The viability of HepG2 and HepG2/cr cells was subsequently measured using the CCK-8 assay in the presence or absence of 8 μM cisplatin, over a time course of 0–72 h. (**G,H**): Chemoresistant cell migration analysis. Cells (400,000/well) were grown in 12-well culture plates. After 24 h, a horizontal wound was created using a sterile 10 μL pipette tip. Cells were then washed (twice) with culture medium and floating cells were removed. Any remaining cells were incubated prior to further observations. (**G**) Images were taken following incubation of HepG2 (upper panel) and HepG2/cr (lower panel) cells for 0, 6, 12, and 24 h, using an inverted phase contrast microscope at ×100 magnification. Yellow lines mark the wound area used for quantification (Scale bar = 200 μm). (**H**) Quantification of wound area filled by the both cell lines. Error bars are presented as mean ± SEM of at least three independent determinations and *p* values are presented as * *p* < 0.05; *** *p* < 0.0005; ns = not significant. Wound area was quantified using ImageJ2.14.0/1.54f software.

Whereas parental cells grew as a distinct monolayer, the continuous treatment of HepG2 cells with 8 μM cisplatin for 4 days caused the HepG2/cr cells to cluster and layer up on one another, producing a colony-like phenotype. The cells that survived a single dose of cisplatin were less sensitive to subsequent cisplatin treatment than their parental counterparts (Figure 1). The colony-like phenotype was formed for those cells that survived cisplatin treatment and their clonal progeny then grew in colony form. It may not necessarily be the case that resistant cells show the colony-like growth pattern after recovery from the flask in which they were challenged to become resistant. They were collected by trypsinisation and grew similarly to non-resistant cells, but were less sensitive to the cisplatin; they became resistant (Figure 1B,C). This change was characterised by an increase in $IC_{50}$ from parental levels of 8 μM (following 24 h cisplatin exposure) to 19 μM in chemoresistant cells (Figure 1D). Following 48 h retreatment, the $IC_{50}$ of parental HepG2 cells dropped to 4 μM, while that of HepG2/Cr cells remained at approximately 19 μM (Figure 1E). Interestingly, both parental and HepG2/cr grew at similar rates following subsequent incubation (Figure 1F), an effect not seen in scratch test assays (Figure 1G). Chemoresistant cells were used for subsequent experiments between passages numbers 1–3.

Unlike measurements made directly on cells grown on plates, scratch test migration assays indicated that the initial growth rate of cisplatin-resistant cells was approximately 60% faster than that of parental cells (Figure 1G). Cisplatin-resistant cells covered the wound area within 24 h, whereas parental cells took longer to achieve the same effect (Figure 1G,H).

### 3.2. TG2 Isoform Expression and Transaminating Activity Modulated in HepG2/cr Cell Lines

Whereas the total mRNA levels of TG2 in HepG2/cr were approximately 50% of those of parental cells, this reflected changes in TG2-L and not in TG2-S protein levels, which stayed approximately constant (Figure 2A,B). Interestingly, the TG2's enzymatic transamidation activity in HepG2/cr cells was greater than that of their parental counterparts, though not enough to be statistically significant (Figure 2C).

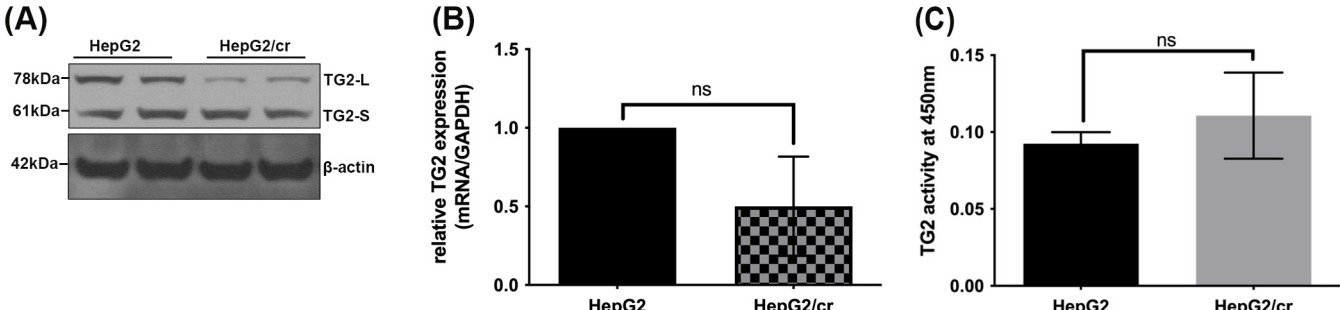

**Figure 2.** A comparison of TG2 expression in parental HepG2 and cisplatin-resistant HepG2/cr cells. (**A**) Total cell protein was extracted from the lysates of equal numbers ($1 \times 10^6$ cells) of HepG2 and HepG2/cr cells, using RIPA buffer. Western blot analysis of TG2 protein expression was then performed using an anti-mouse TG2 polyclonal antibody (lanes represent duplicate samples for each cell line). Other experiments showed the same general pattern of results, although there was minor TG2-S movement from the membranous fractions in the chemoresistant cells after cisplatin treatment in the other two experiments. (**B**) Total TG2 mRNA expression was measured using RT-PCR. (**C**) Transaminase activity was measured using a TG2-specific colorimetric assay kit. TG2 activity is the enzymatic activity of TG2 to perform protein–protein cross linking, transamination, and deamidation of proteins. Error bars are presented as mean $\pm$ SEM of at least three independent determinations and *p* values are presented as *p* = 0.05 (**B**) and *p* > 0.05 (**C**). ns = not significant.

### 3.3. Alexa Fluor 546-Labelled Cisplatin Uptake by HepG2 and HepG2/cr Cell Lines

Following the treatment of HepG2 cells for 1 h, a finely granular pattern of fluorescence became distributed throughout the cytoplasm and nucleus in the majority of parental HepG2 cells (Figure 3A). In comparison, in HepG2/cr cells, the Alexa fluor 546 appeared to accumulate more selectively around the nuclear membrane and/or its closely opposing cytoplasm and did not appear to enter the nuclei (Figure 3A). When the cellular uptake of Alexa fluor 546-labelled cisplatin was measured using flow cytometry, we observed that the cisplatin-resistant HepG2/cr cells took up less fluorescence-tagged cisplatin compared to parental cells (Figure 3B).

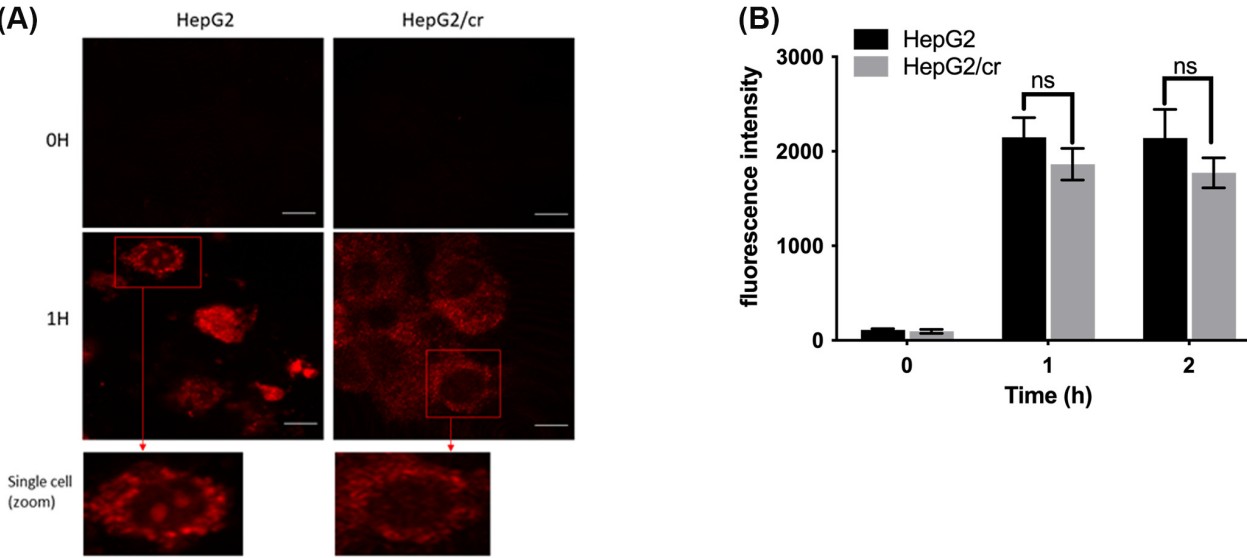

**Figure 3.** Fluorescence distribution and uptake of Alexa fluor 546-labelled cisplatin by HepG2 and HepG2/cr cells. (**A**) HepG2 cells compared against single-step-selected, cisplatin-resistant HepG2/cr

cells. Cells were observed using confocal microscopy at ×400 magnification, with excitation at 543 nm (scale bar = 10 μm). Results are representative of triplicate experiments. (**B**) Cellular uptake of Alexa fluor 546-labelled cisplatin measured using flow cytometry. Approximately $5 \times 10^5$ cells were incubated with Alexa fluor 546-labelled cisplatin, at a final concentration of 40 U/mL, in 6-well culture plates for 0, 1, and 2 h. Then, cells were collected and washed with PBS, before finally being re-suspended in ice-cold PBS. Cells (10,000) were analysed immediately using flow cytometry. Error bars are presented as mean $\pm$ SEM of duplicate samples from three independent determinations and $p$ values are presented as $p > 0.05$; ns = not significant.

### 3.4. Effects of Cystamine on the Uptake of Alexa Fluor 546-Labelled Cisplatin

Cystamine treatment (0–2 mM) alone had little or no effect on cell viability [45]. The results show that TG2 transamidation activity was slightly higher (though not significantly so) in HepG2/cr cells compared to that of parental cells (Figures 2C and 4A). The very modest inhibition of TG2 activity achieved following 48 h treatment (Figure 4A) was accompanied by a similar increase in the intracellular accumulation or uptake of Alexa fluor 546-labelled cisplatin in both HepG2 and HepG2/cr cell lines; the effect was greater in parental cells than in chemoresistant cells (Figure 4B).

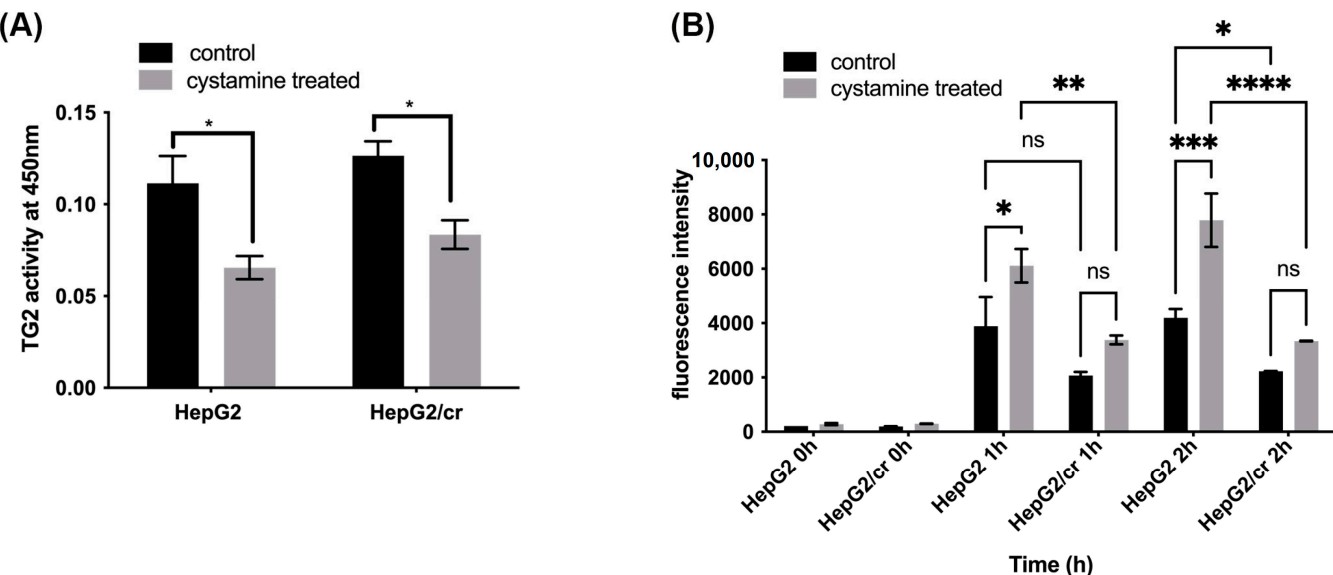

**Figure 4.** Inhibition of TG2 activity increased the uptake of fluorescent cisplatin in both parental HepG2 and chemoresistant HepG2/cr cell lines. (**A**) The parental HepG2 and cisplatin-resistant HepG2/cr cells were treated or untreated (control) with the TG2 inhibitor cystamine, at a final concentration of 2 mM for 48 h, while TG2 activity was measured using a TG2-specific colorimetric assay kit. (**B**) The cystamine pre-treated and untreated cells were incubated with Alexa fluor 546-labelled cisplatin at a final concentration of 40 U/mL for 0, 1, or 2 h. Cells were then collected using centrifugation, washed with PBS, and their fluorescence intensity determined using flow cytometry. Error bars are presented as mean $\pm$ SEM of duplicate samples from three independent determinations. The statistical significance of results against $p$ values was analysed using two-way ANOVA with Sidak's multiple comparison test. * $p < 0.05$; ** $p < 0.01$; *** $p < 0.001$; **** $p < 0.0001$; ns = not significant.

### 3.5. Effects of siRNA Silencing of TG2 Expression on the Uptake of Cisplatin

Inhibition of TG2 gene expression by mRNA silencing (siRNA) was effective at eliminating almost all the TG2-L isoform expression in both parental and chemoresistant cells. Although most of the TG2-S was also removed, both cell lines retained traces of TG2-S following silencing (Figure 5A, lanes 2 and 4); control treatments with carrier alone had minimal effects on cell viability.

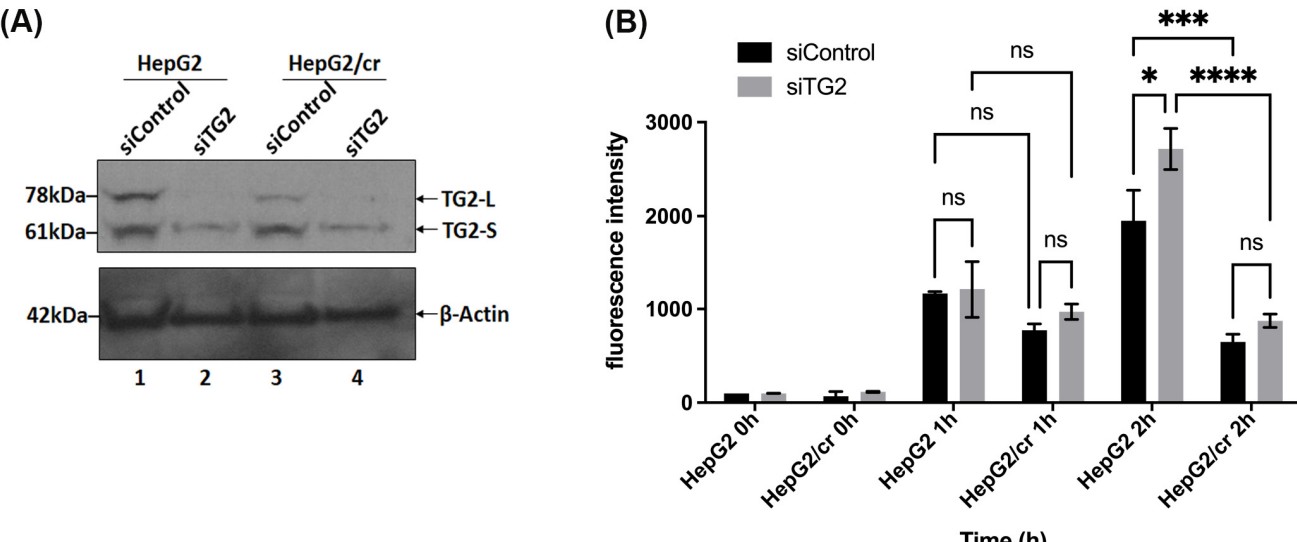

**Figure 5.** Cellular uptake of Alexa fluor 546-labelled cisplatin after TG2 mRNA silencing, as measured using flow cytometry. (**A**) Western blot showing reduction in TG2 expression, after siRNA treatment for 48 h. (**B**) Cellular uptake of fluorescent cisplatin after siRNA treatment. Both parental and resistant cells were treated or not treated with siRNA specific for TG2 mRNA for 48 h; then, cells were incubated with Alexa fluor 546-labelled cisplatin at a final concentration of 40 U/mL, for up to 2 h. The labelled cells were then trypsinised and harvested, before suspension in ice-cold PBS. Fluorescence intensities were subsequently measured using flow cytometry. The results are means $\pm$ SD of duplicate samples from three independent experiments. The statistical significance of results against $p$ values was analysed using two-way ANOVA with Sidak's multiple comparison test. * $p < 0.05$; *** $p < 0.001$; **** $p < 0.0001$; ns = not significant.

Following TG2 silencing treatment, the cellular uptake of Alexa fluor 546-labelled cisplatin increased in a time-dependent manner, with a more pronounced effect in parental cells compared to chemoresistant cells; the chemoresistant cells took up less than half that of parental cells, following 2 h treatment (Figure 5B). The pattern of protein isoform expression showed that TG2-L silencing increased cisplatin uptake in both HepG2 and HepG2/cr cell lines, as the fluorescence intensity was greater in siRNA-treated cells compared to non-treated (control groups) in both cell lines, with the overall pattern of effect being similar to cystamine treatment. In addition, following TG2 silencing treatment, Alexa fluor 546-labelled cisplatin was clearly visible in the nucleus of both HepG2 and HepG2/cr cell lines, whereas the fluorescence signal was barely detectable in the non-silenced HepG2/cr control group (Figure 6A).

*3.6. Changes in Intracellular Distribution of TG2-L and TG2-S Accompany Chemoresistance*

When untreated cytoplasmic and membrane-bound extracts of parental and chemoresistant cells were compared using Western blot analysis, marked differences in the subcellular distribution of the two predominant isoforms of TG2 were observed in both cases, whereas almost all the protein products of the long transcript of TG2 (TG2-L) were present in the cytoplasm of parental cells; in stark contrast, almost all of the products of the short transcript (TG2-S) were membrane-bound. However, this distribution dramatically changed in response to the treatment of parental cells with 8 $\mu$M cisplatin for 24 h. Following this treatment of non-resistant cells, TG2-S redistributed almost entirely from its membrane-bound form to a non-membrane-bound, cytoplasmic form (Figure 6B). The dramatic nature of this change that accompanied the cellular toxicity of cisplatin to parental cells was not seen in cisplatin-treated chemoresistant HepG2/cr cells, where toxicity effects were greatly reduced and where TG2-S remained almost entirely membrane-associated (Figure 6C).

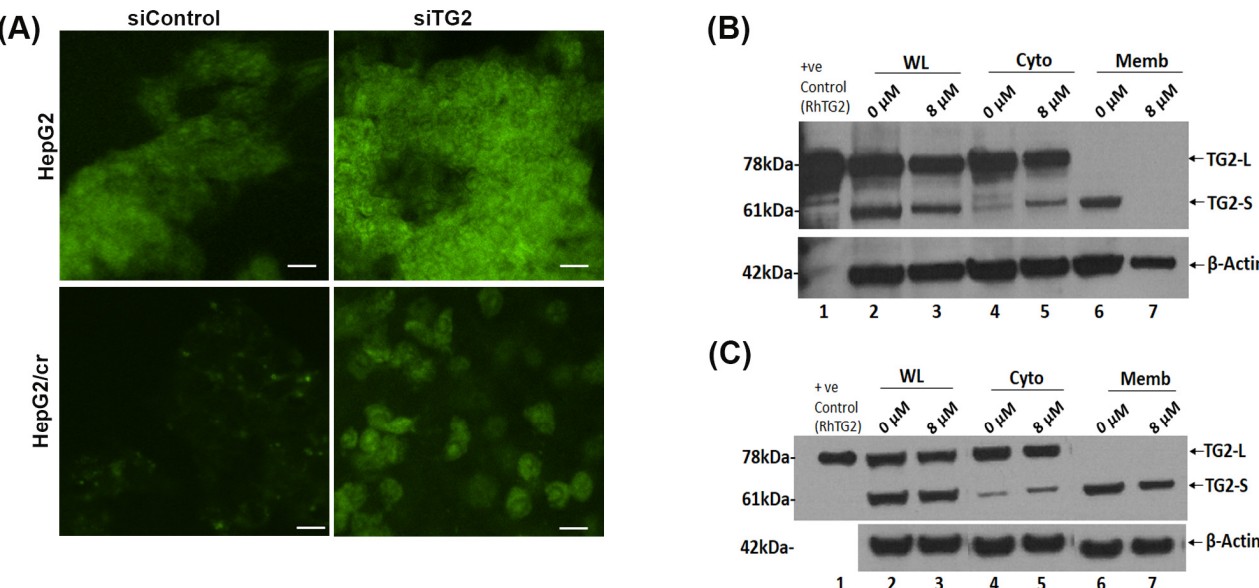

**Figure 6.** (**A**): Cellular uptake of Alexa fluor 546-labelled cisplatin after TG2 siRNA silencing, analysed using confocal microscopy. TG2 expression by HepG2 (upper panel) and HepG2/cr (lower panel) cells was reduced, following treatment with a specific anti-TG2 siRNA oligomer for 48 h; then, cells were incubated with Alexa fluor 546-labelled cisplatin, at a final concentration of 200 U/mL for 1 h. Cells were then fixed in 70% ethanol at −20 °C, before being mounted on slides with fluorescence mounting medium. The slides were observed using confocal microscopy at ×400 magnification, with the excitation at 546 nm (scale bar = 20 μm). (**B,C**): Western blot analysis showed the differential localisation of TG2 isoforms both before and following treatment of HepG2 and HepG2/cr cells with 8 μM cisplatin for 24 h. Cytoplasmic and membrane-associated proteins were extracted with a commercially available membrane preparation kit. (**B**) Parental HepG2; (**C**) chemoresistant HepG2/cr cells. WL: whole lysate; Cyto: cytoplasm; Memb: membrane, RhTG2: commercially available recombinant human TG2 protein control.

### 3.7. Membrane Protein–Protein Interaction and Molecular Docking Simulation of TG2 Isoforms

When the TG2 protein sequence (accession n. P21980) was scanned for potential functional sites using the Eukaryotic Linear Motif (ELM) prediction server, several validated interaction sites and putative short linear motifs (SLiMs) based on the primary amino acid sequence of TG2-L were detected. Amongst these, three short stretches of adjacent amino acids overlapping with each other were identified as the canonical LC3 (microtubule-associated protein 1A/1B-light chain 3)-interacting region (LIR) motif at positions 487 to 500 of TG2. Interestingly, these putative LIR motifs are also present on the C-terminal domain of the TG2-S isoform, where they are exposed at the surface of the protein and may be less susceptible to the structural constraints posed by the additional C-terminal domain present in the full length TG2 (Figure 7). Although many protein–protein interactions are mediated by SLiMs, we decided to investigate this further, to exclude false positive findings and to identify any other potential interaction between TG2 and LC3. The structures of the individual proteins in their free, unbound form were analysed using the computational docking platform HADDOCK, to predict and model the three-dimensional structure of the putative molecular complex (Figure 7). The predicted structure of the TG2-LC3 complex revealed that the two proteins were interacting via an interface region located on the surface of each protein. Detailed analysis of the molecular model complex using PDBsum showed that the region of TG2 involved in the protein–protein interaction was 30 residues long and included the LIR motif identified with ELM. The molecular complex was stabilised by hydrogen bonds, salt bridges, and non-bonded contacts (Figure 7).

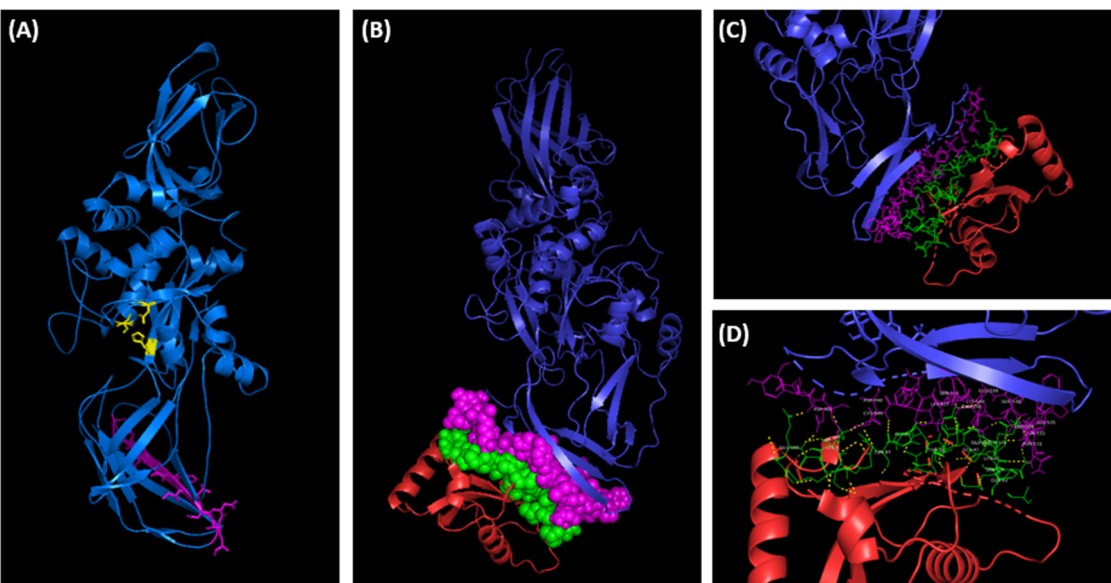

**Figure 7.** (**A**) Three-dimensional structure prediction model of TG2-S (residues 1–548) generated using Phyre2 (version 2.0) protein modelling software [49]. The TG2-S isoform (blue) is predicted to assume the open conformation, exposing the catalytic triad—cysteine277, histidine335, and aspartate358 (yellow)—as well as the functional motif LIR (magenta). The predicted LIR region is located in the C-terminal β-barrel domain B1. (**B**) The protein–protein docking simulation of TG2 (blue) and LC3 (red) generated using HADDOCK (version 2.2). The predicted interacting residues on TG2 (magenta) and LC3 (green) are shown as spheres, to indicate the relatively large contact surfaces between the two proteins. (**C,D**) Cropped images of the molecular complex showing the interactive residues of TG2 (magenta) and LC3 (green). The hydrogen bonds are shown in yellow in (**D**). The images were rendered using PyMOL (version 2.6.0) molecular visualisation system software.

The majority of the residues on TG2 (27) were located on the C-terminal β-barrel domain B1 of TG2-L, with only three residues located on the β-barrel domain B2, thus indicating that the β-barrel domain B1, which is retained in the TG-2 isoform, is the major contributor to the binding and stability of the TG2-LC3 complex. Most of the interactive residues were grouped into a 22 amino acid long linear motif, located at the C-terminus of the β-barrel domain B1 (Figure 7). The 23 amino acid long interactive region of LC3 was located within the exposed flexible loops of the protein, as shown in Figure 7. The interactive residues were arranged in short linear motifs comprising 3–6 amino acids, which are likely to engage in specific, but transient interactions with TG2.

## 4. Discussion

Drug resistance ensues owing to a variety of mechanisms, such as increased drug inactivation, drug efflux from cancer cells, enhanced repair of chemotherapy-induced damage, activation of pro-survival pathways, and inactivation of cell death pathways [50]. For instance, increased glutathione S-transferase (GST) expression has been associated with resistance to cisplatin-based chemotherapy [51]. The conjugation of cisplatin to endogenous glutathione (GSH) by GST has been reported to mediate the extrusion of the GSH conjugate via specialised transmembrane efflux pumps, such as the multidrug resistance-associated protein 2 (MRP2) [50]. However, the GHS-mediated detoxification process appears to be impaired in HepG2 cells, where the expression or the catalytic activity of GST has been shown to be relatively low [52,53], thus suggesting that alternative mechanisms may be responsible for chemoresistance. We postulate that the membrane-associated TG2-S isoform is directly or indirectly involved in the dynamic autophagy pathway that can gradually lead to decreased cisplatin cytotoxicity in HepG2-Cr cells, thus promoting cell survival.

Although it is widely accepted that high levels of TG2 expression in neoplastic hepatocellular carcinoma cells contribute to an increased risk of chemoresistance [45,54,55], the reasons for this are still unclear. We believe that the results presented here clarify the mechanistic role(s) of TG2 in drug resistance and provide further evidence for the role of TG2 in cellular processes, resulting from the integration of cellular stress and death signals.

The results presented here show that HepG2 cells exposed to cisplatin quickly developed resistance to the drug. This unwanted side effect is seen in patients and the magnitude of the increase in $IC_{50}$, from 4 to 19 μM, in chemoresistant cells after 48 h places this model system firmly within the clinically relevant range. We saw that single-dose exposed chemoresistant cells migrated faster when plated on 2D culture; it would be pertinent to investigate in the future whether a similar phenomenon is observed in 3D culture and in in vivo experiments such as those using xenograft models. As expected, fluorescence measurements of labelled cisplatin showed that chemoresistant cells took up less fluorescently labelled cisplatin than non-chemoresistant cells, suggesting that lower cellular uptake and, thus, restriction of intracellular concentrations of cisplatin are important factors and are possibly the key reason for the development of the chemoresistant phenotype in these cells.

We have reported previously that, when normal (non-chemoresistant) cultured HepG2 cells are toxified by cisplatin, they typically lose 30–50% of both their TG2-L expression and transamidating activity; this could account for an increase in uptake of cisplatin with a consequent increase in drug toxicity and a loss of metastatic potential [45]. We therefore expected that the development of chemoresistance might be associated with a greater total TG2 expression compared to normal parental cells and that TG2 overexpression would be accompanied by a decreased intracellular uptake of cisplatin, as this seems the most likely basis for the development of the chemoresistant phenotype [16].

Counterintuitively, however, we found that, rather than increasing TG2 expression, the development of chemoresistance to cisplatin in hepatocellular carcinoma cells was accompanied by a marginal decline in the levels of total TG2 mRNA and protein, compared to parental cells (Figure 2B), though the TG2 enzymatic activity was relatively unaffected and may even have slightly increased (Figure 2C). Moreover, unlike the previous study [45], where the antibody used was only capable of measuring TG2-L expression, the antibody used in the current study proved to be cross-reactive to both TG2-L and TG2-S isoforms, confirming that TG2-S levels were also unchanged during the development of chemoresistance (Figure 2A).

Despite the lack of change in the expression of TG2 protein during development of chemoresistance, support for a TG2 role in the modulation of cisplatin uptake came from two sources. Firstly, by specifically downregulating gene expression of TG2 using anti-TG2 siRNA, it was possible to largely eliminate TG2 protein expression, without causing significant toxic effects to these cells. This is in line with the evidence that gene knock-out animals are capable of survival [27] and showed that the specific targeting of TG2 expression was accompanied by a large increase in cisplatin uptake in both parental and chemoresistant cells. Secondly, the treatment of cells with the non-toxic broad-range TG2 enzyme inhibitor 2 mM cystamine—a treatment that is likely to inhibit both TG2 isoforms—showed a similar pattern of effects to that of TG2 silencing (Figure 4), where an overall 40% loss of TG2 activity correlated with almost a doubling of cisplatin uptake after 2 h of treatment, an effect that was almost twice as great in parental cells compared to their chemoresistant counterparts.

Although confirming a role for TG2 in chemoresistance, the lack of difference in TG2 expression and catalytic activity between parental and chemoresistant cells was initially puzzling and it was collectively suggested that, if TG2 is indeed an important factor in restricting cisplatin uptake during chemoresistance, then this effect must be conferred by some form of post-translational compensatory mechanism. For instance, it cannot be ruled out that some form of activation, such as the activation of the cellular pool of TG2, may contribute to the survival of chemoresistant cells, following re-exposure to cisplatin, a process presumably not fully operational in the non-chemoresistant parental cells. Only

following the subcellular isolation of lysates obtained from parental and chemoresistant cells, did it become apparent that it was not the differential expression of TG2 isoforms that was responsible for the development of cisplatin chemoresistance but it was, in fact, the differential localisation of TG2 isoforms that appeared to confer the TG2-mediated chemoresistant effects. The data obtained in this study showed that TG2-L appears to be cytoplasmically distributed in the cell, whereas most of the TG2-S appears to be associated with subcellular membranous structures. The release of the membrane-associated TG2-S in the cytosol that occurred in response to cisplatin toxicity in parental HepG2 cells appears to mediate cell death, an effect not seen in the chemoresistant cells, where the TG2-S was retained in the membrane fraction (Figure 6).

If this were the case for parental HepG2 cells, then it might reasonably follow that, in contrast, anything that favours the retention of TG2-S in its membrane-sequestered form might also suppress apoptosis, thus delaying cell death and promoting cell survival and chemoresistance [56,57]. In this scenario, the TG2-dependent blocking effect on drug uptake and cell death would be expected to persist in chemoresistant cells, as re-treatment with cisplatin has little effect on TG2-S membrane association (Figure 6C). As a result, calcium-dependent levels of cross-linking may be lower in chemoresistant cells compared to parental cells, resulting in the inhibition of apoptosis and, thus, promoting survival. Hence, the intracellular location and activity state of transglutaminase differentially impacts on cell death and the role of TG2 in cellular functions [57–61].

### 4.1. Translocation of TG2 from the Cytoplasm to the ECM May Contribute to Chemoresistance

We cannot say, at this point, whether the membrane association of TG2-S is mediated by membrane-associated protein–protein interactions, or whether it operates via direct interaction with membrane lipids of the plasma membrane, or indeed other membranous structures in cells. However, it is perhaps of interest that TG2 is known to associate with plasma membrane proximal exosomes that are released by cancer cells in response to stress [57] and which may contribute to fibrosis—a common pathological sequelae to hepatocellular carcinoma.

The translocation of cytosolic TG2 (TG2-L) from the cytoplasm to the extracellular space takes place via the fusion of exosomes with the outer cell membrane. It has been shown that the transamidation activity of TG2 results in protease-resistant intermolecular or intramolecular isopeptide bonds that effectively crosslink extracellular matrix (ECM) fibrils, stiffening the ECM and, thus, protecting the cell from the cytotoxic damage exerted by cisplatin [62]. This is in line with the observation that resistance to the anti-cancer drug doxorubicin in breast cancer cells seems to be dependent upon the action of TG2 on ECM proteins, thereby promoting the interaction between integrins and fibronectin [56]. The unblocking of drug uptake using inhibitors of the TG2 enzyme activity may, therefore, offer a strategy for increasing the susceptibility of cancer cells to chemotherapeutic drugs, in order to induce apoptosis in cancer cells. Interestingly, as previously reported, cystamine derivatives are clinically well tolerated and so may present a route to the cisplatin/cystamine co-treatment of patients with hepatocellular carcinoma, to overcome or limit the development of chemoresistance [45,61,63].

### 4.2. TG2-S' Role in Cell Survival and Chemoresistance following Cisplatin Treatment

The uptake of cisplatin caused the majority of sensitive HepG2 cells (90%) to undergo apoptosis, with a small proportion of cells surviving cisplatin cytotoxicity and acquiring resistance to the drug. In these cells, TG2-S was found to be retained in the membranous fraction, thus suggesting that the localisation of this protein in sub-cellular compartments is a critical factor in determining drug resistance and cell survival. Future suggested work includes more detailed compartmental analysis to understand TG2 subcellular localisation in even greater detail.

In a normal stress-free environment, TG2 is maintained in the closed conformation state, acting as a G-protein inside the cell and only rarely is TG2 found to adopt a cat-

alytically active open conformation, as this enzyme requires levels of $Ca^{2+}$ well above the physiological range for its activation [64]. The TG2-S splice variant lacks the C-terminal GTP-binding regulatory domain, which controls the response of TG2 to $Ca^{2+}$ activation. As a consequence, TG2-S displays a deregulated transamidation activity in the cytosol, because of the predominant role of the desensitisation to GTP. The absence of the C-terminal domain causes TG2-S to adopt a conformation in solution, resembling the open state of the full-length TG2 that facilitates LC3 binding, as shown in Figure 7 [65,66].

Although the association of TG2 with membrane structures is not completely understood, several studies have advanced the idea that TG2 can interact with specific membrane-associated proteins during autophagy. For example, the interaction of TG2 with the membrane-bound microtubule-associated protein light chain 3 (LC3 II) via the autophagy adaptor protein p62 has been shown to facilitate the cross-linking of misfolded protein inside the autophagosome [67–69]. In a separate study, TG2 bound to the p62-LC3 heterodimer protein complex was shown to associate with the tumour protein 53 (p53) in the autophagosome, thus suggesting that TG2 may function as a molecular chaperone and facilitate the translocation of 'client' proteins to the autophagosome [70]. Given that TG2 protein contains several functional regions [33], it would not be surprising to find that several other 'client' proteins could be recruited by the ternary TG2 complex in the same way, for transfer to the autophagy-mediated degradation pathway.

The association of TG2 with the autophagosome membrane has been reported to occur through protein–protein interactions between the C-terminal domain of TG2 and the N-terminal domain of p62 [70]. As the C-terminal domain of TG2 is absent in TG2-S, any potential interaction with the p62-LC3 II heterodimer protein complex must be ruled out. In our study, however, it appears that TG2 may be able to interact with LC3 through the C-terminal 27-residue region located on the β-barrel domain B1, which is present on TG2-L and TG2-S, as well as multiple short linear motifs harboured on the exposed flexible loops of LC3 (Figure 7). Hence, it is possible that TG2-S may associate with the autophagosome membrane by directly interacting with LC3/LC3 homologues or other autophagy-related proteins, without the requirement for an additional adaptor protein. Similarly, it is tempting to speculate that TG2-S may also function as a molecular chaperone and facilitate the recruitment of a diverse range of 'client' proteins for delivery to the autophagosome (Figure 8, left panel). The association of TG2-S with LC3/LC3 homologues could also explain the mechanism by which TG2-S is retained in its membrane-sequestered form in cisplatin-treated chemoresistant HepG2/cr cells.

TG2 is also involved in the homeostasis of the actin cytoskeleton, the regulation of which is essential for proper intracellular trafficking of autophagic vesicles and their fusion with lysosomes [71,72]. In fact, TG2 can post-translationally modify the major components of the cytoskeleton network, such as tubulin, vimentin, and actin, and significantly influence its dynamics [18,19]. Thus, it seems plausible to speculate that the catalytically competent TG2-S isoform could act at the interface between the actin cytoskeleton and the autophagosome, by interacting with key cytoskeleton regulators and LC3/LC3 homologues situated on the outside of the vesicle membrane (Figure 8, left panel). While the actin cytoskeleton has been reported to be involved in cisplatin chemoresistance by physically disrupting the drug uptake [73], mounting evidence supports the view that copper homeostasis proteins ATP7A and ATP7B are directly involved in the intracellular sequestration and transport of cisplatin outside the cell [74]. Unlike ATP7A, which is ubiquitously expressed in various cells and tissues, ATP7B has a more limited expression pattern, with the highest expression level in the liver. It has recently been reported that ATP7B co-localises with LC3 in HepG2 cells, suggesting that copper and, presumably, cisplatin clearance could be mediated by autophagy [74]. Hence, TG2-S could directly or indirectly contribute to cisplatin detoxification, by facilitating the intracellular trafficking of autophagosomes and their fusion with endolysosomal vesicles by modulating the actin cytoskeleton dynamics (Figure 8, right panel).

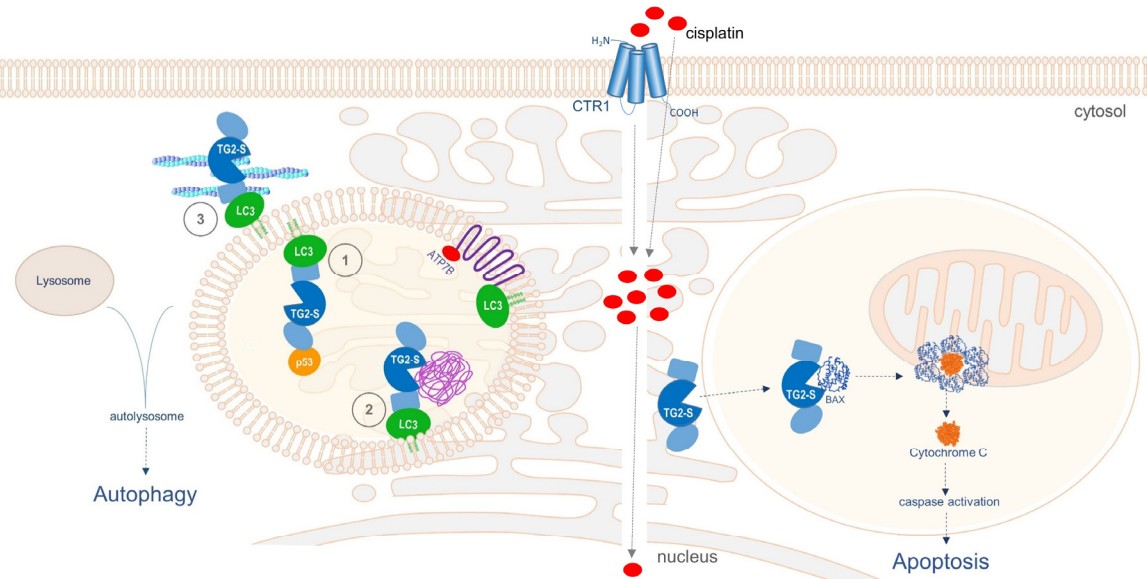

**Figure 8.** Diagram illustrating the postulated dual role of TG2-S, following cisplatin uptake in chemoresistant HepG2/cr cells (**left panel**) and parental HepG2 cells (**right panel**). The cellular uptake of cisplatin has been widely reported to be mediated by several membrane transporters, such as the copper transporter 1 (CTR1) and by passive diffusion [50]. Left panel: TG2-S may participate in multiple cellular events such as (1) recruitment and chaperoning of client proteins, such as p53 or BAX, for delivery to the autophagosome; (2) cross-linking of protein cargo and aggregate formation during autophagy; and (3) assisting the intracellular trafficking of autophagosomes and their fusion with endolysosomal vesicles by modulating the actin cytoskeleton dynamics. Right panel: TG2-S may be involved in BAX-induced Cytochrome C release and caspase activation, leading to apoptosis. The domains of TG2-S (in blue) are the core domain (central circle), the N-terminal domain (oval shape) and the C-terminal domain (rectangular shape). The aggresome is depicted in purple.

### 4.3. TG2-S' Role in Apoptic Cell Death following Cisplatin Treatment

It is possible that the expression or the post-transcriptional regulation of the membrane scaffold protein LC3/LC3 homologue is/are affected by multiple cytotoxicity events, which have the effect of hampering or weakening the interaction between TG2-S and LC3/LC3 homologues.

Most cells undergo apoptosis through the intrinsic pathway. This is dependent on mitochondrial outer membrane permeabilisation, which is mediated by the pro-apoptotic Bcl-2 family proteins, BAX and BAK [75]. During apoptosis, BAX translocates from the cytosol to the outer mitochondrial membrane, wherein it contributes to the formation of pores to release cytochrome *C*. The analysis of the TG2 primary sequence shows the presence of an eight amino acid domain, highly homologous to the Bcl-2 family BH3 domain [76]. It has been postulated that TG2-BH3 can directly interact with BAX and direct cell death via pore formation on the mitochondrial outer membranes, triggering the release of cytochrome C in the cytosol. This process requires the presence of a catalytically active open form of TG2 that allows free access to the catalytic triad—cysteine277, histidine335, and aspartate358—responsible for transamidating activity, while simultaneously allowing access to the BH3 domain, located at position 204–211 of the catalytic core of TG2 [76]. We suggest that cisplatin-induced DNA damage triggers the release of the TG2-S isoform to the cytosol of cisplatin-sensitive HepG2 cells.

Therefore, in addition to its transamidation-competent open conformation, TG2-S would also allow for the interaction between the BH3 domain and the cytosolic BAX monomers, a process which is essential for the insertion of BAX oligomers in the mitochondrial membrane and the consequent oligomerisation of BAX proteins [75,76]. The release of cytochrome *C* through BAX oligomeric pores formed in the mitochondrial membrane is the

key event initiating the apoptotic cascade (Figure 8). This mechanism appears to unravel, at least in part, the puzzle posed by a pro-survival protein inducing cell death, when held in its open state.

To summarise, the evidence presented in this study indicates that the sub-cellular localisation of TG2 isoforms is a major factor in the regulation of its various biochemical and physiological activities, which subsequently trigger diverse downstream events. This indicates that the pro-apoptotic or anti-apoptotic effects of TG2-S are fundamentally dependent on its cellular context and structural conformation. Our observations also support the contention that there may be potential for using TG2 levels and isoform distribution as biomarkers in optimising cisplatin dose responses, especially if cisplatin efficiency is enhanced by its combination with relatively non-toxic doses of specific TG2 inhibitors, such as DON compounds (6-diazo-5-oxo-L-norleucine-containing peptides) or other recently developed TG2 inhibitor small peptides [77]. The results presented herein also add a new dimension to the increasingly established role for TG2-S, rather than the TG2-L isoform, as the arbiter of cell death, but, in this case, only following its release from a membrane-bound pool. It would be interesting to validate the hypothetical interaction of TG2-S with LC3/other proteins in in vitro and in vivo studies.

**Author Contributions:** All the experimental work described in this manuscript was carried out by D.D.M.; the protein structural analysis was carried out by C.F. and P.J.C.; and C.V.S.P. contributed to the conceptualisation of the research framework; the experimental design of the work; and critically read, edited, and commented on the manuscript. All authors have read and agreed to the published version of the manuscript.

**Funding:** This research received no external funding.

**Institutional Review Board Statement:** Not applicable.

**Informed Consent Statement:** Not applicable.

**Data Availability Statement:** Any supporting data can be made available upon request.

**Acknowledgments:** The authors would like to thank Social Welfare of Maharashtra State, Social Justice and Special Assistance Department, Government of Maharashtra State, India, for sponsoring the PhD scholarship of D. D. Meshram (Resolution No. EBC-2010/C.No.174/Edu-1). All other contributions of equipment and chemicals were funded or supplied by Anglia Ruskin University, Cambridge, UK.

**Conflicts of Interest:** The authors declare no conflicts of interest.

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
