# Peer review of "Membrane Association of the Short Transglutaminase Type 2 Splice Variant (TG2-S) Modulates Cisplatin Resistance in a Human Hepatocellular Carcinoma (HepG2) Cell Line"

_cimb, doi:10.3390/cimb46050259_

Round 1

Reviewer 1 Report

Comments and Suggestions for Authors

In this study, the authors found that deficient subcellular relocalisation of TG2-S from membranous structures into the cytoplasm may limit the apoptic response to cisplatin toxicity in chemoresistant cells. Moreover, structural analysis of TG2 revealed the presence of binding motifs for interaction of TG-S with the membrane scaffold protein LC3/LC3 homologue that could contribute to a novel mechanism of chemotherapeutic resistance in HepG2 cells. There are several comments to address before meeting the criteria for publication.

1. The authors conducted the experiments in vitro in this study. However, experiments in vivo are also crucial to validate the results. Please discuss or add xenograft results into the experiments.

2. Some of the figures are compressed and distorted. Please check and redo the picture.

3. In Figure 2A, the healing areas of the cells were not clear enough. Please mark the frontiers of the would healing area in the pictures.

4. In Figure 3A, the gray scale of the background of WB is not consistent among the blots. Please check the exposure title or readjust the pictures.

5. The font size in Figure 5 is not consistent. Please check.

6. Some of the figures could be merged to make the pictures more concise.

7. I would encourage the authors to further demonstrate the results and molecular mechanism in a graphical abstract to make it easier to understand.

8. The authors demonstrated that structural analysis of TG2 revealed the presence of binding motifs for interaction of TG-S with the membrane scaffold protein LC3/LC3 homologue that could contribute to a novel mechanism of chemotherapeutic resistance in HepG2 cells. But they did not validate this via experimental results.

Comments on the Quality of English Language

Not applicable.

Reviewer 2 Report

Comments and Suggestions for Authors

Meshram et al. are attempting to show that the treatment of HepG2 cells with cisplatin results in a shift of TG2-S from a membrane-associated complex to a cytoplasmic pool, an effect that was not observed in cells that had been made resistant to cisplatin. While the results and experiments appear to support this hypothesis, the overall manuscript is challenging to comprehend due to the authors either providing excessive detail or insufficient information. Additionally, some figures lack statistical analysis and controls. Furthermore, some conclusions seem to be drawn too hastily.

Comments:

·      Authors should mention the current treatments available for patients. As immune-checkpoints inhibitors are now used as first line treatment.

·      Could authors explain why cisplatin is re-emerging as a source of treatment?

·      Did authors try to seed the HepG2r cells on collagen-treated plate? This may render these cells capable to plate in 2D and not 3D.

·      Can authors explain figure 1D, why cells start at 150% and 80% of viability? It makes the results difficult to understand. Shouldn’t it be 100%?

·      Can authors explain the differences between Fig 1D and E? Do cells proliferate after the most sensitive ones died? Do they have any evidence?

·      Figure 1 and 2 can be combined. 

·      On the scratch test, resistant cells don’t form colony-like structure. Why?

·      Figure 3: A and B should be inverted. Can authors quantify the bands and show more samples by western blot. N=5 would be great. Authors should explain what is this TG2 activity.

·      Figures 4 and 7 are missing nuclei staining (Dapi/Hoechst). In Figure A, the staining observed in the nucleus could be an artefact staining (nucleoli). Does cisplatin is known to enter the nucleus?

·      Authors cannot claim: “When the cellular uptake of Alexa fluor 546-labelled cisplatin was measured by flow cytometry, the fluorescence intensity was lower in cisplatin-resistant HepG2/cr cells compared to parental cells, though not enough to be statistically significant (Figure 4B)”.

·      Authors should show these data: “Cystamine treatment (0-2 mM) alone had little or no effect on cell viability (data not shown)”.

·      What is the effect of cystamine on the 2 isoforms of TG2? 

·      Figure 6B lacks statistical analysis between HepG2 and HepG2cr. They cannot draw any conclusion without that.

·      The cytonucleus western blot lacks all compartment controls.

Round 2

Reviewer 1 Report

Comments and Suggestions for Authors

The authors have provided satisfactory response to the comments and I do not have further suggestions.

Comments on the Quality of English Language

The authors have provided satisfactory response to the comments and I do not have further suggestions.

Reviewer 2 Report

Comments and Suggestions for Authors The authors have significantly enhanced their paper. While some experiments may not seem particularly robust to me, the narrative and scientific discoveries remain pertinent.